# Effects of Nickel Nanoparticles on *Rhodococcus* Cell Surface Morphology and Nanomechanical Properties

**DOI:** 10.3390/nano12060951

**Published:** 2022-03-14

**Authors:** Maria S. Kuyukina, Grigorii G. Glebov, Irena B. Ivshina

**Affiliations:** 1Microbiology and Immunology Department, Perm State University, 614990 Perm, Russia; grisha899@mail.ru (G.G.G.); ivshina@iegm.ru (I.B.I.); 2Institute of Ecology and Genetics of Microorganisms, Perm Federal Research Center, Russian Academy of Sciences, 614081 Perm, Russia

**Keywords:** nickel nanoparticles, actinobacteria, *Rhodococcus*, atomic force microscopy, elasticity, adhesion, zeta potential, hydrophobicity

## Abstract

Nickel nanoparticles (NPs) are used for soil remediation and wastewater treatment due to their high adsorption capacity against complex organic pollutants. However, despite the growing use of nickel NPs, their toxicological towards environmental bacteria have not been sufficiently studied. Actinobacteria of the genus *Rhodococcus* are valuable bioremediation agents degrading a range of harmful and recalcitrant chemicals. Both positive and negative effects of metal ions and NPs on the biodegradation of organic pollutants by *Rhodococcus* were revealed, however, the mechanisms of such interactions, in addition to direct toxic effects, remain unclear. In the present work, the influence of nickel NPs on the viability, surface topology and nanomechanical properties of *Rhodococcus* cells have been studied. Bacterial adaptations to high (up to 1.0 g/L) concentrations of nickel NPs during prolonged (24 and 48 h) exposure were detected using combined confocal laser scanning and atomic force microscopy. Incubation with nickel NPs resulted in a 1.25–1.5-fold increase in the relative surface area and roughness, changes in cellular charge and adhesion characteristics, as well as a 2–8-fold decrease in the Young’s modulus of *Rhodococcus ruber* IEGM 231 cells. Presumably, the treatment of rhodococcal cells with sublethal concentrations (0.01–0.1 g/L) of nickel NPs facilitates the colonization of surfaces, which is important in the production of immobilized biocatalysts based on whole bacterial cells adsorbed on solid carriers. Based on the data obtained, cell surface functionalizing with NPs is possible to enhance adhesive and catalytic properties of bacteria suitable for environmental applications.

## 1. Introduction

Nanoscale nickel particles with a large specific surface area, magnetic and catalytic properties are used in the chemical and electronics industries, as well as for wastewater treatment due to their high adsorption capacity against complex organic pollutants [1]. However, despite the growing use and accumulation of nickel nanoparticles (NPs) in the environment, the toxicological aspects of their effects on living organisms, in particular bacterial cells, have not been sufficiently studied [2,3].

The interaction of nanometals with bacterial cells is a complex and multidirectional process, varying from antibacterial action, manifested in the formation of reactive oxygen species (ROS) and damage to cell membranes [4], inhibition or activation of growth, mobility and chemotaxis [5,6], changes in the zeta potential and elastic–mechanical properties of cells, increasing their roughness [7,8] to the possibility of nanomodification of the cell surface to enhance functional activity [9].

Actinobacteria of the genus *Rhodococcus* degrade a number of harmful and persistent chemicals, such as petroleum hydrocarbons, phenols, solvents, pesticides and pharmaceutical pollutants and, therefore, are used in bioremediation [10,11]. They are characterized by resistance to adverse environmental factors and accumulation of metal salts [11,12], which determines their use in environmental biotechnologies. Rhodococci are able to sequester heavy metal ions by biosorption and active accumulation; However, the study of their interaction with metal nanoparticles is only at the very beginning [11,12,13]. Further research is needed to reveal particular effects of nanometals on the physiology and catalytic properties of *Rhodococcus*. A number of studies have shown both positive and negative effects of metal ions and NPs on the biodegradation of organic pollutants by rhodococci [13,14,15]; however, the mechanisms of such effects, in addition to the direct toxic effects of NPs on bacterial cells, remain unclear.

In this work, the complex effects of nickel nanoparticles on the viability, morphology, cellular charge, hydrophobicity and nanomechanical properties of *Rhodococcus* cells are revealed. Despite the toxic effects at high concentrations, nickel NPs are shown to enhance surface properties related to bacterial adhesion and consumption of hydrophobic substrates. Such nanonickel-activated *Rhodococcus* species can be used in various fields of environmental nanobiotechnology.

## 2. Materials and Methods

### 2.1. Nickel Nanoparticles

Nickel nanoparticles suspended in water and stabilized with β-cyclodextrin were purchased from the M9 Company (Tolyatti, Russia). Suspensions of Ni NPs were centrifuged at 3000 rpm for 15 min, washed with sterile 10 mM KNO_3_ and sonicated for 2 min using Soniprep 150 (MSE, London, UK) immediately before use. The average particles size in the stock nanoparticle suspension was in the range from 50 to 140 nm as measured by dynamic light scattering analysis using a ZetaSizer Nano ZS (Malvern Instruments, Malvern, UK). Ni NPs were used in tenfold concentrations (0.0001–1.0 g/L). A ZetaSizer Nano ZS analyzer (Malvern Instruments, UK) was used to measure the zeta potential of nickel NPs and bacterial cells by electrophoretic light scattering.

### 2.2. Bacterial Strains and Culture Conditions

Bacterial strains used were *Rhodococcus opacus* IEGM 263, *Rhodococcus rhodochrous* IEGM 1363 and *Rhodococcus ruber* IEGM 231 from the Regional Specialized Collection of Alkanotrophic Microorganisms, Perm, Russia (IEGM; www.iegmcol.ru; WFCC-WDCM 768; UNU/CKP 73559/480868). Bacteria grown in nutrient broth (28 °C, 3 days) were harvested, washed twice and suspended in ddH_2_O or 10 mM KNO_3_ to an optical density (OD_600_) of 0.5. The relative hydrophobicity of bacterial cells was determined in salt aggregation test (SAT) using ammonium sulfate in increasing concentrations of 0.2, 0.6 and 1.0 M [16]. Ten microliters of bacterial suspension in ddH_2_O was gently mixed with 10 μL of (NH_4_)_2_SO_4_ on the microscope slide. Cell aggregation was recorded by phase contrast microscopy (Axiostar Plus, (Zeiss, Jena, Germany)) using a 100× immersion lens (numerical aperture 1.4). According to this method, the more hydrophobic the bacterial cell wall, the lower the salt concentration at which cell aggregation was observed. A minimum of 3 fields of each variant were examined.

### 2.3. Combined Confocal Laser and Atomic Force Scanning

A combined system consisting of a confocal laser scanning microscope (CLSM) Olympus FV1000 (Olympus, Tokyo, Japan) and an atomic force microscope (AFM) Asylum MFP-3D-BIOTM (Asylum Research, Santa Barbara, CA, USA) was used as described earlier [17]. Bacterial cells were stained with the LIVE/DEAD^®^ BacLightTM Bacterial Viability Kit (Invitrogen, Waltham, MA, USA) to determine living and dead cells and scanned in a CLSM using a 100x immersion lens (numerical aperture 1.4). Image analysis was performed with the FV10-ASW 3.1 software package (Olympus, Tokyo, Japan). Then the CLSM images were imported into the AFM software (Igor Pro 6.22A, WaveMetrics, Tigard, OR, USA) and AFM scanning of the desired area was performed in air, in tapping mode, using aluminum-coated silicon cantilevers AC240TS (Olympus, Tokyo, Japan) with a stiffness coefficient of 2 N/m, a radius of curvature of the needle of 7 nm, a resonant frequency of 70 kHz. The dimensions (length and width) and the root-mean-square (RMS) roughness of the cell surface were calculated using the signals from the height channel. A minimum of 20 cells of each variant were scanned and calculated.

### 2.4. Nanomechanical Properties Using AFM Spectroscopy

AFM spectroscopy of bacterial cells was performed in a Fluid Cell Lite (Asylum Research, Santa Barbara, CA, USA) in contact mode using the iDrive magnetic excitation system for scanning biological samples in liquid. Chromium- and gold-coated silicon nitride cantilevers TR400PB (Olympus, Tokyo, Japan) with a stiffness coefficient of 0.09 N/m, a radius of curvature of the needle of 30 nm and a resonant frequency of 32 kHz were used. Force mapping was carried out with a resolution of 32 × 32 pixels, with each pixel of the force map corresponding to a separate section of the cell surface for which the force curve was constructed. The analysis of force maps, masking of the image (of the bacterial cell from the rest of the image), calculation of the adhesion force and Young’s modulus of elasticity were carried out in the IGOR Pro 6.22A software package (WaveMetrics, Tigard, OR, USA). A minimum of 5 cells of each variant were scanned and calculated.

### 2.5. Statistical Analysis

Experimental data were statistically analyzed using a standard Excel program, calculating the mean and standard deviation (m ± SD), shown in figures as SD bars.

## 3. Results and Discussion

### 3.1. Effects of Nickel NPs on Viability and Hydrophobic Properties of Rhodococcus Cells

According to the CLSM data (Figure 1), the viability of *Rhodococcus* strains decreased as a result of incubation with Ni NPs, depending on the concentration of nanoparticles and incubation time. In particular, 48-h exposure of *R. rhodochrous* IEGM 1363 to a maximum (1.0 g/L) concentration of Ni NPs resulted in 100% cell death, while the presence of 68–100% and 36–64% of viable cells at low (<0.01 g/L) and medium (>0.01 g/L) concentrations of NPs indicates moderate resistance of this strain to nanonickel.

The *R. ruber* IEGM 231 cells showed a similar tendency to decrease viability with an increase in the concentration of Ni NPs but were characterized by higher resistance (14% of viable cells) to the maximum (1.0 g/L) concentration of nanoparticles. The largest (23%) number of living cells after 48 h of exposure with 1.0 g/L of Ni NPs was recorded for *R. opacus* IEGM 263. The viability of these two strains was not affected significantly by low (<0.01 g/L) concentrations of nanonickel (average viability rates of 77–100%). Treatment with medium (0.01–0.1 g/L) concentrations of NPs resulted in 55–70% of cell survival. Such differences in rhodococcal resistance to metal NPs can be explained by their physiological peculiarities. It was previously shown [12] that representatives of *R. ruber* and *R. rhodochrous* synthesizing red-orange non-diffusing pigment were more resistant to heavy metal ions, in particular Ni^2+^, compared with non-pigmented *Rhodococcus* species.

Lipid components of the bacterial cell wall determine its degree of hydrophobicity, which, in turn, contributes to the cell aggregation under adverse environmental conditions [11,18]. The obtained SAT test results (Figure 2) indicate that three studied *Rhodococcus* strains were characterized by high cell hydrophobicity (cell aggregation was recorded at 0.2 M ammonium sulfate). Treatment with increasing concentrations of Ni NPs did not significantly affect the degree of hydrophobicity of rhodococcal cells. The data obtained are consistent with previous studies [11,16] confirming the hydrophobic nature of *Rhodococcus* cells as their intrinsic adaptation to the assimilation of hydrocarbon substrates.

### 3.2. Effects of Nickel NPs on Zeta Potentials of Rhodococcus Cells

The electrokinetic (zeta) potential is an integral parameter of the bacterial surface charge and a sensitive indicator of damage to the cell membrane when exposed to toxic metal nanoparticles [7,8]. When exposed to Ni NPs, *R. opacus* IEGM 263 cells exhibited a shift (by 3.5–7 mV) of the control negative value of the zeta potential further into the negative region (Figure 3). In particular, each 10-fold increase in the concentration of NPs caused a monotonic (with the exception of 0.01 g/L) increase in the negative charge of *R. opacus* cells, which was further attenuated at the maximum (1.0 g/L) concentration. At the same time, only a slight (by 1–2 mV) increase in the zeta potential modulus of *R. ruber* IEGM 231 and *R. rhodochrous* IEGM 1363 cells was recorded upon the contact with 0.0001–0.1 g/L of Ni NPs, which shifted to a less negative region at the maximum (1.0 g/L) nanometal concentration. Previously, a significant increase in the negative charge of *Staphylococcus aureus* and, to a lesser extent, *Escherichia coli* cells incubated with silver bionanoparticles clearly correlated with a decrease in cell viability, indicating a varied antibacterial effect of Ag NPs against Gram-positive and Gram-negative bacteria [8]. In the present study, different trajectories of zeta potential changes during treatment with increasing concentrations of Ni NPs indicate species or strain differences in the surface structures of *Rhodococcus* cells, causing their different resistance to nanometals.

### 3.3. Effects of Nickel NPs on the Cell Size and Surface Roughness

Combined CLSM-AFM scanning revealed the concentration-dependent effects of Ni NPs on the morphology of *Rhodococcus* cells. As an example, deformations of the normal cell shape, accompanied by pronounced inhomogeneity of the cell surface are shown for *R. ruber* IEGM 231 cells incubated for 48 h with 1.0 g/L of Ni NPs (Figure 4), thus confirming the toxicity of high concentrations of the nanometal.

An important morphometric characteristic is the specific surface area of a bacterial cell, defined as the ratio of its surface area (S) to volume (V) and characterizing the relative contact area of the cell with environmental factors [19]. After 24 h of incubation with low (0.0001–0.001 g/L) concentrations of Ni NPs, *R. ruber* IEGM 231 cells became thinner and decreased in size, thereby increasing by 1.25–1.5 times the specific contact surface with the nanometal (Table 1). At the same time, at high (0.1–1.0 g/L) concentrations of Ni NPs, an increase in cell size was accompanied by a 10% decrease in the cell area-to-volume ratio (S/V), which persisted at 48 h of incubation (with the exception of 1.0 g/L Ni NPs). The revealed tendency to decrease the specific surface area of cells in contact with nanometal was even more pronounced for the Ni-sensitive strain of *R. rhodochrous* IEGM 1363. In this case, an increase in the S/V index was observed only at a minimum (0.0001 g/L) concentration and incubation time (24 h) with Ni NPs, and in all other cases, an increase in size and a decrease (by 15–48%) in the specific surface area of cells were recorded, especially noticeable during prolonged (48 h) contact with the nanometal. These data and previously reported [20] similar changes in the specific surface area of *R. ruber* IEGM 231 cells (an increase in contact with less toxic solvents—toluene and cyclohexane and a decrease in cell contact with ethanol) indicate universal morphological adaptations of rhodococci to different toxic factors [11].

The surface ultrastructure, in particular, the roughness of the cell wall, or a complex of surface irregularities forming a unique microrelief of bacterial cells, is primarily transformed when exposed to damaging factors [21]. To further study the stress response of *Rhodococcus* to high concentrations of Ni NPs, the AFM profiles of cells were analyzed, and RMS surface roughness values were calculated (Figure 5). It was found that control bacterial cells were characterized by a smoother surface compared to the nanoparticle-treated cells. Previously [8], an increase in the roughness of bacterial cells was also observed as a result of the membranotropic action of silver nanoparticles. As expected, the toxic effect of Ni NPs on rhodococci was higher after 48 h of exposure compared with daily incubation. Interestingly, lower concentrations (0.0001–0.01 g/L) of Ni NPs caused a maximum increase in the surface roughness (by 26 and 34% after 24 and 48 h, respectively) of *R. ruber* IEGM 231 cells, while higher (0.1–1.0 g/L) concentrations of nanometals led to a lesser (by 16 and 13%, respectively) change in cell surface relief.

The cells of *R. opacus* IEGM 263, initially having a smoother surface, when incubated with lower (0.0001–0.01 g/L) concentrations of Ni NPs, showed relatively stable values of RMS roughness, slightly deviating from the control figures. A further increase in the nanoparticle concentration to 0.1–1.0 g/L led to maximum changes in the *R. opacus* microrelief (by 37 and 42% after 24 and 48 h, respectively). The RMS roughness of *R. rhodochrous* IEGM 1363 cells increased more significantly than *R. ruber* IEGM 231 and *R. opacus* IEGM 263, especially during 48-h incubation with Ni NPs, thus indicating a stronger damaging effect of the nanometal on this strain. The maximum RMS roughness values of *R. rhodochrous* IEGM 1363 cells were observed in the presence of 0.0001 and 1.0 g/L of Ni NPs, suggesting the absence of a strict concentration dependence of cell wall damage during prolonged exposure to the nanometal.

### 3.4. Effects of Nickel NPs on Nanomechanical Properties

Nanomechanical properties of *R. ruber* IEGM 231 cells after their incubation with different concentrations of Ni NPs were studied using AFM force spectroscopy in liquid (Figure 6). The obtained force maps indicated an increase in the adhesion force between the cantilever probe and the surface of the bacterial cell under the influence of Ni NPs, which correlated (R = 0.9968, *p* < 0.05) with an increase in the RMS roughness of cells.

Apparently, the rough cell surface facilitates the attachment of rhodococci to a solid substrate due to the exposure of highly adhesive sites of the cell wall [11]. Consequently, the treatment of *Rhodococcus* cells with Ni NPs will contribute to more efficient colonization of surfaces, which is important, for example, in the production of immobilized biocatalysts based on bacterial cells adsorbed on solid carriers.

It is generally accepted [22] that an increase in the Young’s modulus, as a measure of the rigidity or elasticity of the cell wall, leads to a decrease in the adhesion efficiency of bacterial cells. However, in another study [7], treatment with hematite nanoparticles increased the rigidity of *E. coli* cells and caused the appearance of many adhesive sites on the cell surface. Unlike *E. coli* cells, which initially had no detectable adhesion peaks (adhesion force of 20–100 pN), native *R. ruber* IEGM 231 cells were characterized by relatively high adhesion force (0.41 nN) and Young’s modulus (49 MPa) values (Table 2). Upon the treatment with nanonickel, a Young’s modulus of *R. ruber* IEGM 231 cells decreased proportionally to the concentration at 0.01–0.1 g/L of Ni NPs, and at a maximum concentration of 1.0 g/L it doubled.

A decrease in the Young’s modulus when treating cells with metal nanoparticles suggested a weakening of rigid bonds between cell wall components, the appearance of softer areas, leading to damage to the bacterial cell wall, peptidoglycan layer and cytoplasmic membrane, including leakage of intracellular contents [8]. On the other hand, a less rigid and more flexible hydrophobic cell wall contributed to the uptake of hydrocarbon substrates by *Rhodococcus* cells [20]. Contradictory literature data on the influence of nanometals on mechanical and rheological properties of bacterial cells are partially explained by technical variations in the AFM nanoindentation and structural heterogeneity of the bacterial cell surface, represented by macromolecules with varying degrees of hydrophobicity, electric charge and spatial organization, causing heterogeneity of rheological properties [7,22]. The specific molecular interactions that cause these mechanical forces are the result of a complex interplay of fundamental hydrophobic, electrostatic, van der Waals and hydrogen bonding forces [23]. Nevertheless, the revealed adaptation features, such as cell hydrophobicity accompanied by the increased specific surface area and roughness, an increase in adhesion force and a decrease in Young’s modulus in response to nickel NPs, would determine *Rhodococcus*’s efficient attachment to solid surfaces and degradation of hydrocarbon substrates [11,13,14,22]. Further research on the mechanisms of bacterial adaptation to metal NPs, such as changes in surface morphology (size and roughness) and nanomechanical properties will help to strengthen the advantages of *Rhodococcus* in biotechnology and environmental applications.

## 4. Conclusions

In this study, the concentration-dependent effects of nickel NPs on the viability, surface topology and mechanical properties of *Rhodococcus* cells suggested the possibility of their nanomodification with selected nanometal concentrations in order to improve adhesive and hydrocarbon-degrading activities. It has been proposed that these properties, important for bacterial survival in the environment and the performance of biocatalytic functions, can be modulated by sublethal concentrations of nickel NPs. However, to mitigate detrimental impacts of high concentrations of nickel NPs on *Rhodococcus*, selection of resistant strains or adaptation of bacteria using a laboratory evolution approach would be required.

## Figures and Tables

**Figure 1 nanomaterials-12-00951-f001:**
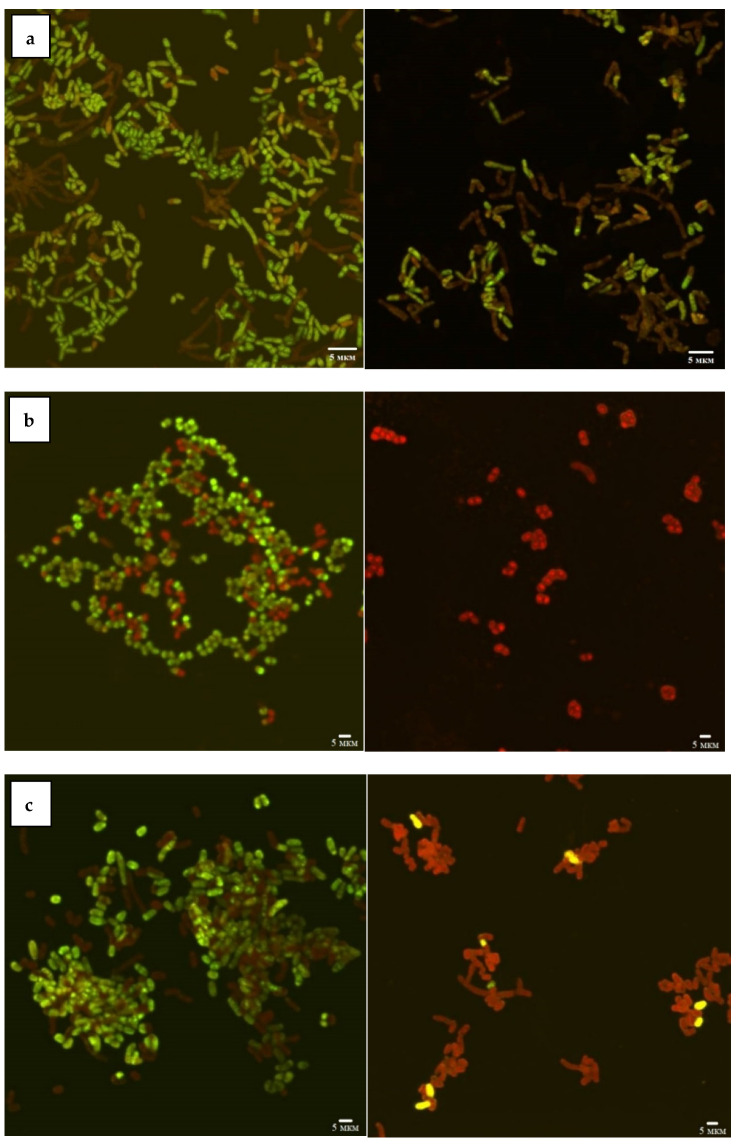
CLSM images of *R. opacus* IEGM 263 (**a**), *R. rhodochrous* IEGM 1363 (**b**), *R. ruber* IEGM 231 (**c**) control cells (**left**) and after 48-h incubation with 1.0 g/L Ni NPs (**right**). Viable cells fluoresce green, dead cells red.

**Figure 2 nanomaterials-12-00951-f002:**
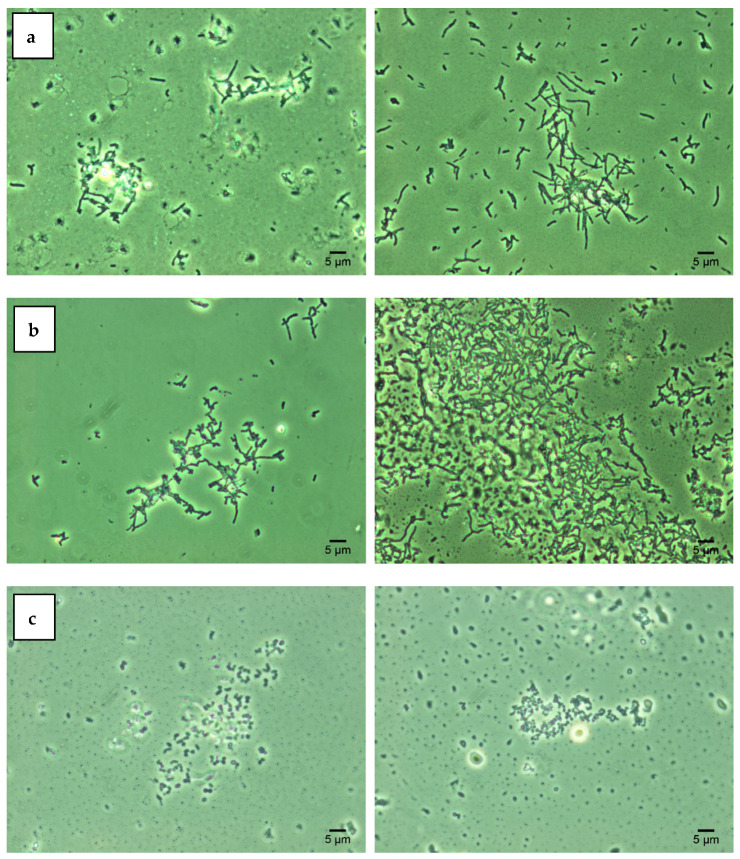
Autoaggregation of *R. opacus* IEGM 263 (**a**), *R. rhodochrous* IEGM 1363 (**b**), *R. ruber* IEGM 231 (**c**) cells in the presence of 0.2 M ammonium sulfate (SAT test): controls (**left**) and after 48-h incubation with 1.0 g/L Ni NPs (**right**). Phase-contrast microscopy × 1000.

**Figure 3 nanomaterials-12-00951-f003:**
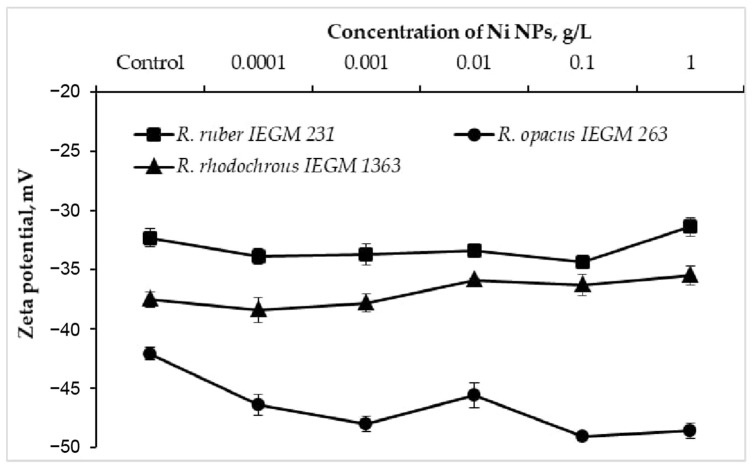
Zeta potential values of *Rhodococcus* cells incubated with different concentrations of Ni NPs.

**Figure 4 nanomaterials-12-00951-f004:**
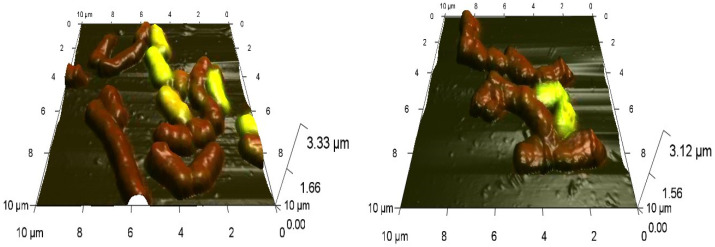
Combined CLSM-AFM images of *R. ruber* IEGM 231 cells in control (**left**) and after 48-h incubation with 1.0 g/L Ni NPs (**right**). Viable cells fluoresce green, dead cells red.

**Figure 5 nanomaterials-12-00951-f005:**
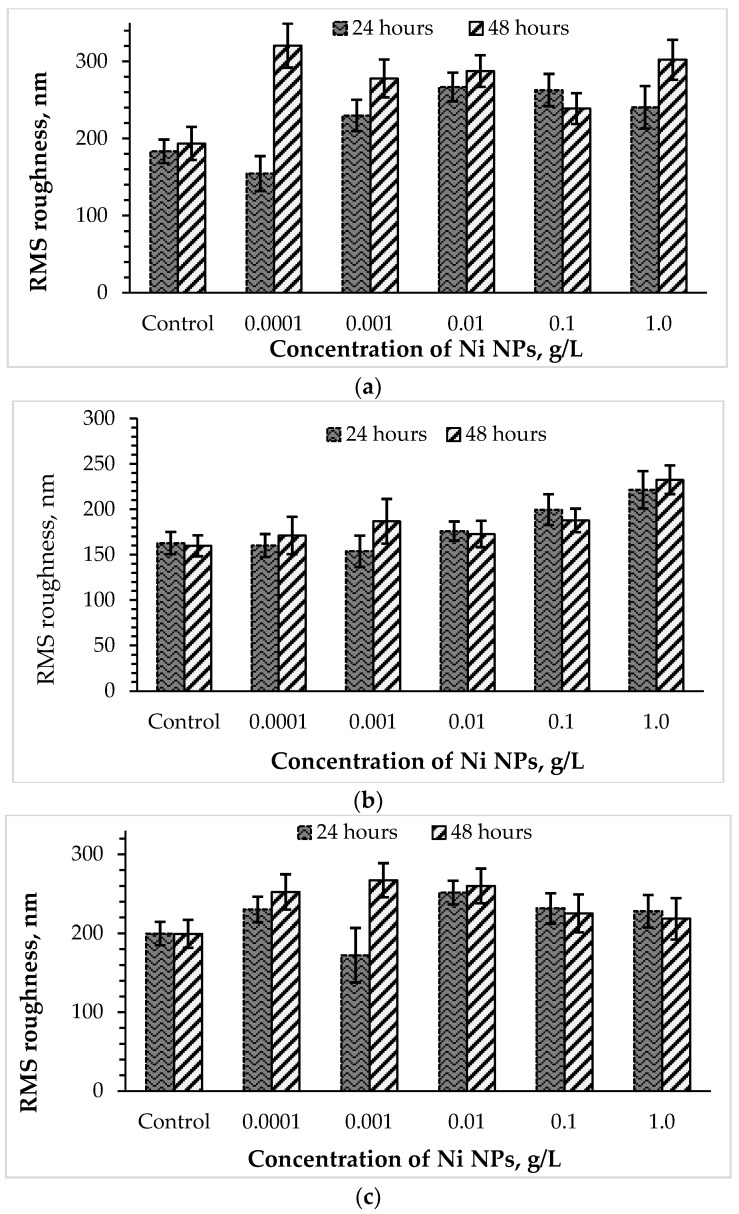
Changes in the surface roughness of *R. opacus* IEGM 263 (**a**), *R. rhodochrous* IEGM 1363 (**b**) and *R. ruber* IEGM 231 (**c**) cells incubated with different concentrations of Ni NPs.

**Figure 6 nanomaterials-12-00951-f006:**
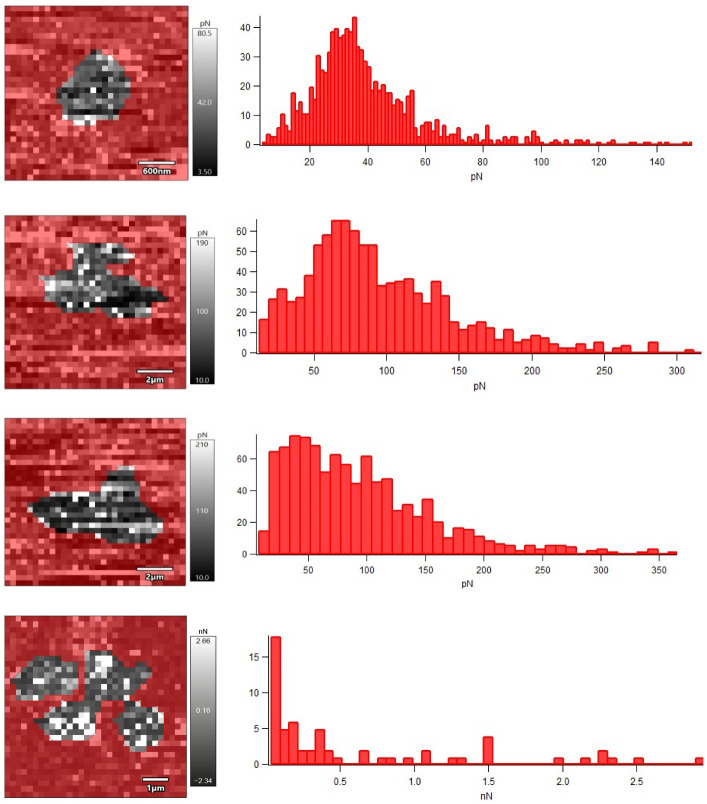
Force maps and histograms of the adhesion force distribution of *R. ruber* IEGM 231 cells incubated with Ni NPs. From top to bottom: control, 0.01, 0.1, 1.0 g/L Ni NPs.

**Table 1 nanomaterials-12-00951-t001:** Effects of Ni NPs on *Rhodococcus* cell dimensions.

Concentration of Ni NPs, g/L	Length, μm	Width, μm	Surface (S), μm^2^	Volume (V), μm^3^	Surface/Volume (S/V)
*R. ruber* IEGM 231					
Control	1.8 ± 0.2 ^a^	0.74 ± 0.08	5.0 ± 0.3	0.8 ± 0.08	6.5 ± 0.5
	1.8 ± 0.1 ^b^	0.74 ± 0.09	4.9 ± 0.4	0.8 ± 0.07	6.6 ± 0.5
0.0001	2.0 ± 0.3	0.57 ± 0.05	4.0 ± 0.4	0.5 ± 0.05	8.1 ± 0.7
	1.4 ± 0.2	0.70 ± 0.06	3.9 ± 0.4	0.5 ± 0.06	7.1 ± 0.6
0.001	2.0 ± 0.2	0.47 ± 0.02	3.3 ± 0.5	0.3 ± 0.05	9.6 ± 0.7
	1.3 ± 0.3	0.85 ± 0.07	4.6 ± 0.4	0.7 ± 0.06	6.2 ± 0.8
0.01	1.9 ± 0.3	0.73 ± 0.06	5.3 ± 0.5	0.8 ± 0.07	6.5 ± 0.5
	1.8 ± 0.2	0.78 ± 0.09	5.3 ± 0.6	0.9 ± 0.07	6.2 ± 0.4
0.1	2.2 ± 0.4	0.82 ± 0.07	6.6 ± 0.7	1.1 ± 0.09	5.8 ± 0.5
	1.8 ± 0.2	0.85 ± 0.09	6.0 ± 0.8	1.0 ± 0.11	5.8 ± 0.7
1.0	1.9 ± 0.3	0.83 ± 0.06	6.1 ± 0.6	1.0 ± 0.08	5.8 ± 0.4
	1.2 ± 0.1	0.59 ± 0.04	2.8 ± 0.3	0.3 ± 0.04	8.4 ± 0.6
*R. rhodochrous* IEGM 1363					
Control	1.0 ± 0.1	0.58 ± 0.06	2.4 ± 0.2	0.3 ± 0.04	8.8 ± 0.9
	1.0 ± 0.2	0.58 ± 0.03	2.4 ± 0.1	0.3 ± 0.03	8.9 ± 0.6
0.0001	1.1 ± 0.2	0.50 ± 0.04	2.1 ± 0.1	0.2 ± 0.01	9.8 ± 0.7
	1.2 ± 0.2	0.58 ± 0.07	2.7 ± 0.3	0.3 ± 0.02	8.5 ± 0.8
0.001	1.1 ± 0.1	0.70 ± 0.05	3.3 ± 0.3	0.4 ± 0.05	7.5 ± 0.7
	1.2 ± 0.2	0.88 ± 0.09	4.4 ± 0.4	0.7 ± 0.06	6.3 ± 0.7
0.01	1.4 ± 0.1	0.73 ± 0.04	4.0 ± 0.5	0.6 ± 0.07	6.9 ± 0.5
	1.3 ± 0.1	0.90 ± 0.07	4.8 ± 0.4	0.8 ± 0.06	6.0 ± 0.4
0.1	1.1 ± 0.3	0.74 ± 0.02	3.4 ± 0.2	0.5 ± 0.04	7.2 ± 0.6
	1.2 ± 0.1	0.95 ± 0.09	5.0 ± 0.3	0.9 ± 0.06	5.9 ± 0.7
1.0	1.2 ± 0.1	1.23 ± 0.14	3.8 ± 0.3	0.6 ± 0.06	6.9 ± 0.8
	1.4 ± 0.2	0.90 ± 0.11	7.8 ± 0.8	1.7 ± 0.12	4.7 ± 0.5

Cell dimensions (mean ± SD) are shown after ^a^ 24 and ^b^ 48 h of incubation with Ni NPs.

**Table 2 nanomaterials-12-00951-t002:** Nanomechanical properties of *R. ruber* IEGM 231 cells after their 24-h incubation with Ni NPs.

Concentration of Ni NPs, g/L	Adhesion Force, nN	Young’s Modulus, MPa
Control	0.041 ± 0.018	48.8 ± 3.9
0.01	0.10 ± 0.04	20.8 ± 2.7
0.1	0.14 ± 0.05	3.2 ± 0.4
1.0	1.42 ± 0.13	6.0 ± 1.2

## Data Availability

Not applicable.

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
