# Peer review of "Effects of Nickel Nanoparticles on Rhodococcus Cell Surface Morphology and Nanomechanical Properties"

_nanomaterials, 2022, doi:10.3390/nano12060951_

Round 1
Reviewer 1 Report
The authors conducted a study on the influence of nickel NPs on the viability, surface topology and nanomechanical properties of Rhodococcus cells. The topic is interesting, the manuscript as a whole is well-organized, well-written, and updated. I have some suggestion to improve the readership of manuscript.
- I think if you add on title Rhodococcus species cells is much better.
Line 22 and 25: Remove nickelLine 14: change rhodococci to Rhodococci and change it italic.
Line 39: add (ROS) after reactive oxygen species
Line 43-45: Sentence need to rewrite
Line 47-51: Long Sentence and please add Ref.
Line 59-60: This Sentence transfer to Conclusions
Line 84-86: transfer to Results and discussion
Effects of nickel NPs on viability and hydrophobic properties of Rhodococcus cells (You should be able to sum up the data in the table ?).
Figure 2 without labelled?
Further re-search on the mechanisms of bacterial adaptation to metal NPs, such as changes in surface morphology (size and roughness) and nanomechanical properties will help to strengthen the advantages of Rhodococcus in biotechnology and environmental applications. (Transfer to end of Results and discussion)
There is no statistical representation in the manuscript. Why?
Author Response
The authors conducted a study on the influence of nickel NPs on the viability, surface topology and nanomechanical properties of Rhodococcus cells. The topic is interesting, the manuscript as a whole is well-organized, well-written, and updated. I have some suggestion to improve the readership of manuscript.
Answer: We thank the Reviewer for the careful reviewing of our paper and the overall positive opinion.
- I think if you add on title Rhodococcus species cells is much better.
Answer: We added “cell”, so the title change to the Effects of Nickel Nanoparticles on Rhodococcus Cell Surface Morphology and Nanomechanical Properties
Line 22 and 25: Remove nickelLine 14: change rhodococci to Rhodococci and change it italic.
Answer: Line 25: We removed nickel. Line 14: We changed to Rhodococcus.
Line 39: add (ROS) after reactive oxygen species
Answer: (ROS) is added.
Line 43-45: Sentence need to rewrite
Answer: The sentence is rewritten to: Actinobacteria of the genus Rhodococcus degrade a number of harmful and persistent chemicals, such as petroleum hydrocarbons, phenols, solvents, pesticides and pharmaceutical pollutants and, therefore, are used in bioremediation [10,11].
Line 47-51: Long Sentence and please add Ref.
Answer: The sentence is divided into two sentences and the references added.
Line 59-60: This Sentence transfer to Conclusions
Answer: The sentence is transferred to Conclusions.
Line 84-86: transfer to Results and discussion
Answer: Since this sentence (According to this method, the more hydrophobic the bacterial cell wall, the lower the salt concentration at which cell aggregation was observed) explains the method details, we decided to leave it in the Materials and methods.
Effects of nickel NPs on viability and hydrophobic properties of Rhodococcus cells (You should be able to sum up the data in the table ?).
Answer: The quantitative data on viability are described in details in the revised text. The hydrophobicity results represent only one concentration of ammonium sulfate 0.2 M (the lowest concentration at which aggregation was observed for all strains tested), so the table would not be informative. Whereas microscopic images, in addition to the results of viability and aggregation, show particular differences in the cell morphology of different Rhodococcus species in cell populations.
Figure 2 without labelled?
Answer: The labels a, b and c are added. Also, 5 µm bars are added.
Further re-search on the mechanisms of bacterial adaptation to metal NPs, such as changes in surface morphology (size and roughness) and nanomechanical properties will help to strengthen the advantages of Rhodococcus in biotechnology and environmental applications. (Transfer to end of Results and discussion)
Answer: This sentence is transferred to the end of Results and discussion.
There is no statistical representation in the manuscript. Why?
Answer: The statistical analysis method has been added as 2.5. Statistical analysis in the Materials and methods. For each method, a number of replications is added.

Reviewer 2 Report
The authors examined the effect of Ni nanoparticles on Rhodococcus cells via confocal and AFM. Overall the works were well designed, however, the work missed a bit of discussion in this study. Few points should be address ed before it can be accepted for publications:
- The authors used 1g/L (1mg/mL). The authors should examine other concentrations range as well for the cell viability and autoaggregation.
- Fig 3 showed the surface charge of cells after treatment with NPs. However, the cells will be aggregated, which may give misleading data.
- The authors may discuss their mechanical data with other studies: Nanoscale, 2020 Oct 14;12(38):19888-19904; Nature Reviews Methods Primers volume 1, Article number: 63 (2021); Nano Lett. 2021, 21, 18, 7595–7601

Author Response
The authors examined the effect of Ni nanoparticles on Rhodococcus cells via confocal and AFM. Overall the works were well designed, however, the work missed a bit of discussion in this study. Few points should be addressed before it can be accepted for publications:
Answer: We thank the Reviewer for the careful reviewing of our paper and the overall positive opinion.
The authors used 1g/L (1mg/mL). The authors should examine other concentrations range as well for the cell viability and autoaggregation.
Answer: We used Ni NPs in tenfold concentrations (0.0001-1.0 g/l) in the cell viability and autoaggregation experiments, as well as in other experiments. Figures 1 and 2 show the microscopy images at highest concentration (1g/l) of nanoparticles. The effects of lower concentrations are described in details in the revised text.
Fig 3 showed the surface charge of cells after treatment with NPs. However, the cells will be aggregated, which may give misleading data.
Answer: In our study, the aggregation of cells was observed in the presence of salt (ammonium sulfate) in the salt aggregation test (SAT) to characterize cell hydrophobicity. While, Ni nanoparticles even at maximum concentration (1g/L) did not lead to the significant cell aggregation as shown in Fig.1.
The authors may discuss their mechanical data with other studies: Nanoscale, 2020 Oct 14;12(38):19888-19904; Nature Reviews Methods Primers volume 1, Article number: 63 (2021); Nano Lett. 2021, 21, 18, 7595–7601
Answer: In this paper, we focused on the effects of metal nanoparticles on mechanical properties of bacterial cells, so relevant references were used. We thank the Reviewer for suggesting new references and the last reference (a recent review of A. Viljoen with co-authors (2021) was added to discuss the specific molecular interactions that cause these mechanical forces.
